# Transmission of Still Images Using Low-Complexity Analog Joint Source-Channel Coding

**DOI:** 10.3390/s19132932

**Published:** 2019-07-03

**Authors:** Jose Balsa, Tomás Domínguez-Bolaño, Óscar Fresnedo, José A. García-Naya, Luis Castedo

**Affiliations:** Department of Computer Engineering, Universidade da Coruña (University of A Coruña), CITIC Research Center, Elviña, 15071 A Coruña, Spain

**Keywords:** image coding, joint source-channel coding, performance evaluation

## Abstract

An analog joint source-channel coding (JSCC) system designed for the transmission of still images is proposed and its performance is compared to that of two digital alternatives which differ in the source encoding operation: Joint Photographic Experts Group (JPEG) and JPEG without entropy coding (JPEGw/oEC), respectively, both relying on an optimized channel encoder–modulator tandem. Apart from a visual comparison, the figures of merit considered in the assessment are the structural similarity (SSIM) index and the time required to transmit an image through additive white Gaussian noise (AWGN) and Rayleigh channels. This work shows that the proposed analog system exhibits a performance similar to that of the digital scheme based on JPEG compression with a noticeable better visual degradation to the human eye, a lower computational complexity, and a negligible delay. These results confirm the suitability of analog JSCC for the transmission of still images in scenarios with severe constraints on power consumption, computational capabilities, and for real-time applications. For these reasons the proposed system is a good candidate for surveillance systems, low-constrained devices, Internet of things (IoT) applications, etc.

## 1. Introduction

In recent years, there has been an increase in the use of resource constrained devices which transmit multimedia data such as images or video, specially in the context of the Internet of things (IoT). In many applications these systems require reliable, low-complexity and low-latency wireless transmissions [1]. Images, as captured by digital cameras, are composed of a large amount of data, and therefore must be compressed and protected against noise before transmission. The compression and transmission of images has been studied extensively for many years [2,3,4,5]. In spite of their analog nature, still images are usually encoded, stored, and processed in a digital way. First, the camera provides a discrete representation of the scene in the spatial domain by means of a collection of real values (pixels) for each color component. Next, an adequate source encoder is employed to remove the statistical, spatial and perceptual redundancy to obtain an efficient digital representation of the image [6]. There exists a large number of compression methods for still images, although Joint Photographic Experts Group (JPEG) is probably the most popular standard.

In a digital transmission, a channel encoder adds redundancy bits to the source sequence to protect it against channel distortions and the resulting bits are mapped to appropriate signals which are sent over the channel. This approach is based on the source-channel separation principle, which has been shown to be optimal for lossless transmission [7] and lossy compression of analog sources [8]. This strategy greatly simplifies the design of the communication systems since the source encoder and the channel encoder can be optimized separately. However, it requires the use of large blocks at both encoders to approach the theoretical optimal performance, and it can lead to large delay and high computational complexity. On the other hand, the pair of encoders must specifically be designed depending on the channel conditions, e.g., for a particular signal-to-noise ratio (SNR). Whenever this value changes, the source and channel encoders must be updated to the current channel conditions. Hence, an adaptive encoding strategy is needed for time-varying environments in order to track the channel conditions at the receiver and to select the pair of encoders from a large set.

A more general approach consists of jointly designing the source and channel encoders in such manner that the source information is directly transformed into the channel symbols to be transmitted. This approach is known as joint source-channel coding (JSCC) and there exist different JSCC strategies depending on how they try to jointly optimize source and channel codes [9,10,11]. When the objective is to transmit discrete-time continuous-amplitude symbols, analog JSCC has been shown to provide good performance in a large number of scenarios [12,13,14,15,16]. This strategy consists of employing continuous parametric mappings, which efficiently fill up the source space, and presents some advantages with respect to the traditional digital systems based on the source-channel separation principle. First, its computational cost and delay are significantly lower since the encoding operation basically consists of applying a parametric mapping function to each source symbol. Also, it presents a graceful degradation in case of imperfect channel state information (CSI), i.e, if the analog JSCC scheme is optimized considering wrong information about the channel conditions. Finally, these schemes can be easily adapted to time-varying environments by updating the mapping parameters. Conversely, the design of general analog JSCC mappings for different scenarios and arbitrary block sizes is extremely difficult.

Most works addressing analog JSCC focus on the transmission of source symbols which are generated according to a particular statistical distribution [12,17]. Its application in practical scenarios is limited to the experimental evaluation of these techniques over real wireless channels [18], the design of some transmission schemes for images [17,19], and in the context of video broadcasting for bandwidth-constrained channels [20].

Unfortunately, current analog JSCC techniques are not able to efficiently exploit the spatial redundancy when they are applied to the source images directly in the spatial domain. Similarly to the digital processing case, a more clever strategy would consist in encoding and transmitting only the relevant information of the image in the frequency domain. An interesting idea was stated in “Softcast” [21], where a 3D discrete cosine transform (DCT) is applied to a set of frames and the output is grouped into chunks. The chunks with values close to zero are discarded, and the rest is selected to be transmitted using linear analog encoding. In this case, an appropriate selection of the chunks would allow for removing most of the image redundancy with a low impact on the image quality. However, this decision must be made for each chunk of the image. In addition to the chunks, metadata is also sent with a bitmap of the discarded chunks, so that the receiver can reconstruct the image from the received chunks. Hence, this compression strategy generates a large amount of metadata which must be considered for a fair comparison with other schemes. Indeed, metadata must be transmitted to the receiver using a reliable digital encoding scheme which ensures that this information is always received without errors. In this sense, the transmission of metadata is a critical step since the presence of a single error can prevent image reconstruction. An alternative approach to remove the image redundancy, proposed in [22], is to apply compressive sensing to each chunk of frequency coefficients, but the amount of metadata is still considerable. Another interesting scheme to transmit video is proposed in [23], where the video is first encoded with a high efficiency video coding (HEVC) encoder and a 3D-DCT is applied to the residuals (the difference between the original and the encoded video) and the output is transmitted using a Shannon–Kotel’nikov analog mapping.

Note that all the previous works propose hybrid digital–analog schemes, in which digital data is also required to reconstruct the images. In this paper, we address the design of a low-complexity and low-delay analog JSCC scheme for the transmission of still images over wireless channels. Although our proposed system only considers the transmission of individual images, in contrast to the aforementioned works, our system requires negligible metadata to function. To encode an image, its spatial representation is first transformed to the frequency domain using the DCT. After this transformation, the relevant information of each block in the source images is compacted in a few coefficients, and therefore most of the DCT coefficients can be disregarded with a minimum loss in the perceived quality. Next, the set of relevant coefficients are encoded using an analog JSCC mapping and transmitted over the wireless channel with the help of the orthogonal frequency-division multiplexing (OFDM) modulation. We finally compare the performance obtained with the proposed analog JSCC scheme to that of a digital system where the transmitter includes the following blocks: A quantization step, a source encoder, a capacity-achieving channel encoder and a signal modulator. It is worth noting that the procedure of removing the spatial redundancy is based on the same ideas as in previous works, i.e., grouping the DCT coefficients into several sets according to their frequency position, and then disregarding a subset of those, which usually corresponds to the highest frequencies. The main difference resides in the fact that both the grouping and removing strategies are fixed for all the blocks of the images, and they are known at the transmitter and receiver. Hence, the amount of required metadata becomes negligible, although the compression will be less efficient than in other schemes. Moreover, for transmitting similar images we could consider presetting the metadata at both transmitter and receiver, hence obtaining a fully analog system. This system allows for low delay transmissions, due to its low complexity and its analog nature, which means that it does not need to use retransmissions as a digital system in case of errors. Therefore, it is a good candidate for a whole range of applications such as surveillance systems, resource-constrained devices, IoT devices, etc.

### Contributions

The main purpose of this paper is to show that a scheme based on analog JSCC can obtain a performance similar to that of traditional digital systems based on JPEG for the encoding and transmission of still images, providing additional advantages for a large number of scenarios. In particular, the main contributions of the paper are summarized below:The design, optimization and evaluation of an analog JSCC-based scheme for the wireless transmission of images. First, the proposed system exploits the spatial redundancy to lower the amount of image information to be encoded by using a simple and static strategy, which avoids the need of transmitting additional metadata to reconstruct the source image at the receiver. Next, the transmission procedure presents some appealing features such as low complexity and storage requirements, and also negligible delay, which could be useful in the progressive encoding mode. Furthermore, analog JSCC techniques are well known to provide graceful degradation even when the channel information is not accurate enough. Finally, the potential distortions caused by the encoding and transmission operations will be more pleasant for the human visual system than in the case of digital processing, i.e., its actual impact on the perceptual quality will be lower.A fair comparison of the proposed analog JSCC scheme to a closed-loop digital system based on JPEG compression. Such a comparison is focused on evaluating the performance of both schemes in terms of image quality and transmission time. In order to provide more insight in the comparison, we also consider a third alternative transmission scheme which employs a similar strategy to exploit the image correlation as in the analog case, but applies traditional encoding and modulation techniques to transmit the resulting digital data.Finally, an in-depth analysis of the obtained results is carried out to provide a global vision about the suitability of the analog JSCC and digital strategies for the transmission of still images.

## 2. System Model

Let us consider the encoding and transmission of still images using analog JSCC techniques. The performance of the proposed scheme is then compared to that of the traditional digital approaches based on the source–channel separation principle [7]. In that case, the image is first compressed using an adequate source encoder which removes the spatial correlation and, then, the resulting binary sequence is encoded with a capacity-achieving channel encoder. In this paper, we consider the standard JPEG for image coding and turbo codes for channel coding as representative components for the digital systems.

Figure 1 and Figure 2 show the block diagrams corresponding to the transmitter and the receiver, respectively, for the considered communication system. The source image is represented by a matrix of K×K elements corresponding to the pixel values for the luminance component. The values for all the pixels range between 0 and 255, i.e., $m_{i,j}\in \left[0,255\right]\;\forall i,j=1,\ldots ,K$.

After a preprocessing step where we subtract 127 to the pixel values to center the distribution around zero, the DCT is employed to obtain the representation of the source image in the frequency domain. The choice of the DCT is motivated because it is widely applied to image processing and, in particular, it is the transform implemented in the JPEG standard, which we consider as a benchmark. Note that we employ 64-bit floating point numbers for all the calculations which required the use of real-valued numbers.

Following the same idea as in the JPEG standard, the source image is split into blocks of 8×8 pixels and the DCT is individually applied to each block. After this transformation, we obtain another block of 8×8 values, which is stacked into a vector of 64 elements following a zig-zag pattern. Note that the zig-zag scan allows for sorting the resulting DCT coefficients in such a manner that the first element of the vector corresponds to the DC (zero frequency), those following the DC represent slight variations in the spatial domain and correspond to low frequencies and, finally, the last coefficients in the vector represent abrupt transitions in the source image and correspond to high frequencies. The main advantage of using the DCT is that the energy of the image in the frequency domain is generally compacted in a few coefficients corresponding to the lower frequencies. This property allows us to disregard most of the DCT coefficients with a minimum information loss.

The set of vectors with the DCT coefficients is encoded using three different approaches (see Figure 1):A particular analog JSCC encoder based on continuous mappings which directly transforms the real-valued DCT coefficients into the real-valued symbols to be transmitted. Since we aim at designing a low-complexity encoding scheme, the compression operation simply consists of disregarding a fixed number of high-frequency coefficients at each DCT block, as we will explain in the next section.The standard JPEG, which provides an efficient binary representation of images by exploiting the spatial correlation of the original image. The digital sequence is then encoded and modulated using a capacity-achieving channel encoder followed by a quadrature amplitude modulation (QAM) scheme.A digital system, named JPEG without entropy coding (JPEGw/oEC), where the source encoder removes the redundant information in a similar way to that of the analog scheme, thus disregarding a fixed number of DCT coefficients. Next, the resulting data are also encoded using a channel encoder and modulated with a QAM scheme.

At the output of any of these schemes, we will obtain a vector of channel symbols which is transmitted over the channel using OFDM. In particular, we consider an OFDM transceiver with similar parameters to those found in the Long-Term Evolution (LTE) 10 MHz profile, i.e., a sampling frequency of 15.36 Msamples/s, 600 data subcarriers, and a hexagonal pilot pattern. Note that the size of the symbols vector is variable and depends on the scheme employed to encode the source image, thus the length of the OFDM frame will also depend on the size of this vector.

At the receiver, the channel is estimated and equalized using minimum mean squared error (MMSE) algorithms. The OFDM demodulator will hence provide a vector of received symbols after the corresponding channel equalization. These symbols are employed to compute an estimate of the DCT coefficients for each block of 8×8 pixels of the image. Each of these blocks is then converted back to the spatial domain by applying the inverse discrete cosine transform (IDCT). Next, the received image is finally reconstructed from the blocks of pixels at the output of the IDCT.

The quality of the received image is measured according to the structural similarity (SSIM). This metric was proposed to take into account the particular features of the human perceptual system [24]. Hence, the values provided by this metric are in accordance to the perceived quality.

### 2.1. Analog JSCC Scheme

The relevant information of the source images is transmitted to the receiver using analog JSCC techniques. As shown in Figure 1 and Figure 2, the analog JSCC scheme consists of two main blocks: (1) an encoder that uses an analog JSCC mapping to transform the real-valued coefficients at the output of the DCT into the corresponding real-valued channel symbols to be transmitted in an OFDM frame; and (2) a receiver that computes an estimate of the DCT coefficients from the received symbols.

#### 2.1.1. Frequency Block Selection

Another important module at the analog transmitter is the one termed as frequency block selection. As discussed in the introduction, analog JSCC techniques are inefficient to deal with the large spatial redundancy of the images, and it is required to include an additional step to remove the dispensable frequency components and lower the amount of data to be encoded. This module is hence responsible for selecting those DCT coefficients which will be encoded using analog JSCC mappings.

Let us assume that each vector of 64 DCT coefficients is already ordered with the zig-zag scan. As discussed in the previous section, the first elements of these vectors correspond to low frequencies, whereas the last ones capture the information of the images at high frequencies. In general, the probability density function (PDF) of the DCT coefficients is quite different depending on whether they represent low or high frequencies. Moreover, from the point of view of the visual impact, the relevance of the information they carried is also different. Both considerations must be taken into account when designing the analog JSCC scheme specifically for the encoding of images. For this reason, we split each vector of 64 coefficients at the input of the analog encoder into $n_{b}$ variable-size blocks, each one corresponding to different blocks of nearby frequencies.

Figure 3 shows an example of this frequency division applied to the *j*-th block of 8×8 DCT coefficients of the source image, when considering $n_{b}=4$. As observed, the 64 coefficients are first sorted according to their frequency position using a zig-zag scanning pattern. Next, the ordered coefficient vector is split into $n_{b}=4$ different blocks: the first block ($s_{j,1}$) contains only the DC coefficient ($c_{0,0}$), the second block ($s_{j,2}$) groups the three next DCT coefficients ($c_{0,1},c_{1,0},c_{2,0}$), whereas the third ($s_{j,3}$) and fourth ($s_{j,4}$) blocks correspond to medium and high frequencies, respectively. Repeating this operation for each DCT block of the image, we will obtain $n_{b}$ sub-vectors at each step (see Figure 3). Finally, we construct $n_{b}$ vectors, each one comprising the corresponding sub-vectors obtained for all the DCT blocks of the image. Thus, the $n_{b}$ vectors of coefficients corresponding to the source image are obtained as follows
(1)$$s_{i}=\left[s_{1,i},s_{2,i},\ldots s_{T,i}\right]\;\;\;\;\;\;\forall i=1,\ldots ,n_{b},$$
where *T* is the total number of 8×8 blocks which the image is divided into. In this way, each one of the resulting vectors $s_{i}$ contains all the DCT coefficients of the image for a specific range of frequencies. This particular division strategy is intended to group the DCT coefficients with a similar nature and visual importance, and it resembles a multi-resolution decomposition [25] or a compression strategy based on subband coding [26].

Since we are interested in the analog encoding of the information, it is important to imitate the behavior of digital systems, which remove the redundant information of the images by entropy coding. However, the compression rate achieved by analog strategies is rather low because of the difficulty of designing analog mappings for an arbitrary block size. We circumvent this limitation by disregarding the vectors of coefficients corresponding to high frequencies in the DCT decomposition, and whose information is less valuable for the human visual perception system. In particular, $n_{c}$ vectors are selected to be encoded with $1\leq n_{c}\leq n_{b}$, whereas the other $n_{b}-n_{c}$ vectors are disregarded and not considered at the encoding stage. This strategy allows us to significantly reduce the number of coefficients to be encoded with the analog JSCC mappings, whereas its impact on the image quality is minimum. In addition, the low-level complexity of the analog JSCC scheme is preserved. Finally, additional metadata is not required to reconstruct the image since both the transmitter and receiver know beforehand the frequency selection strategy.

#### 2.1.2. Analog Encoder

After the DCT and the frequency selection, we have $n_{c}$ vectors of real-valued coefficients at the input of the analog encoder. Each of these vectors is individually encoded using a given analog JSCC mapping, i.e.,
(2)$$x_{i}=f_{i}\left(s_{i}\right)\;i=1,\ldots ,n_{c},$$
where $s_{i}=\left[s_{i,1},\ldots ,s_{i,N_{i}}\right]$ is the *i*-th vector of DCT coefficients, $N_{i}$ is the number of elements in the *i*-th block, $x_{i}$ is the *i*-th vector of encoded symbols, whose length depends on the mapping function $f_{i}\left(\cdot \right)$. Notice that all the coefficients of the same vector are encoded using the same analog mapping, but different mappings could be applied to different vectors.

In this work, we consider two particular analog JSCC mappings: uncoded transmission, and spherical codes based on the exponentially chirped modulation [27]. The former scheme simply consists of sending a scaled version of the input symbols. In this case, the scale factor is chosen to ensure that the power of the encoded symbols is equal to 1. Thus, the *i*-th vector of encoded symbols is given by $x_{i}=\sqrt{\gamma_{i}}s_{i}$, such as $\left|\right|x_{i}{\left|\right|}^{2}=1$. We also consider analog spherical codes proposed for the bandwidth expansion of the input symbols. Assuming an expansion factor *L*, the mapping function $f_{i}:R\to R^{L}$ generates *L* different encoded versions of each input symbol by using sinusoidal functions with different frequencies. Let $s_{i,j}$ be the *j*-th element of the *i*-th input vector, then its corresponding vector of *L* encoded symbols is given by
$$\begin{array}{cc} x_{i,j}=f_{i}\left(s_{i,j}\right)= & \Delta [\cos\left(2\pi s_{i,j}\right),\sin\left(2\pi s_{i,j}\right),\cos\left(2\pi \alpha s_{i,j}\right),\sin\left(2\pi \alpha s_{i,j}\right),\\ & \;\;\ldots ,\;\cos(2\pi \alpha^{L/2-1}s_{i,j},\sin\left(2\pi \alpha^{L/2-1}s_{i,j}\right)], \end{array}$$
where α is the parameter that determines the mapping shape, and Δ is the normalization factor to ensure that the power of the mapping output is equal to 1. Finally, all the vectors of encoded symbols are stacked into a single vector as follows
(3)$$x_{i}={\left[x_{i,1},\ldots ,x_{i,N_{i}}\right]}^{T},$$
where $N_{i}$ corresponds to the size of the *i*-th input vector. The performance of the analog scheme using this mapping function can be optimized by adapting the value of the parameter α depending on the SNR value. However, the performance loss due to using the same α value for all range of SNR values is not significant, and hence the receiver does not need to feedback to the transmitter information about the channel or the optimal values of the mapping parameter. In the low SNR regime, the optimal value for α is 1 and, therefore, the mapping function converges to a non-linear repetition code. Indeed, we have observed that the same performance can be achieved using the linear version of this code and with a more pleasant distortion for the human visual system.

After encoding the $n_{c}$ vectors of DCT coefficients with one of these two strategies, the resulting vectors are stacked into a single vector which is finally transmitted over the channel using OFDM. The concrete details of the proposed analog scheme, including the parameters of the mappings used, are discussed in Section 3.1.

#### 2.1.3. Analog Decoder

At the receiver, the OFDM demodulator provides a vector with the received symbols which is next employed to compute the estimates of the DCT coefficients. Since the source information consists of real-valued symbols, the optimal decoder is the one which minimizes the mean squared error (MSE) between the original symbols and the estimated ones.

For uncoded transmission, a good approximation of the optimal decoder consists of applying linear MMSE decoding to the received symbols. Nevertheless, in the case of the spherical codes, the mapping function is nonlinear, and the computation of the MMSE estimates requires us to numerically solve the corresponding integrals. Fortunately, the computational cost of the decoding operation can be lowered drastically applying a similar strategy to that proposed for analog JSCC mappings based on the Archimedean spiral [28]. The idea consists of using a low-complexity two-stage receiver which, after the channel equalization, applies maximum likelihood (ML) decoding to the filtered symbols. Denoting ${\hat{x}}_{i}$ as the *i*-th vector of filtered symbols, an estimate of the *j*-th coefficient in $s_{i}$ using ML decoding can be obtained as
(4)$${\hat{s}}_{i,j}=\arg\max_{r}\;\;p\left({\hat{x}}_{i,j}|r\right)=\arg\min_{r}{∥{\hat{x}}_{i,j}-f_{i}\left(r\right)∥}^{2},$$
i.e., the symbol on the source space such that the Euclidean distance between its corresponding encoded vector and the received symbol is minimum. Hence, the ML decoding can be implemented in an efficient way using a low complexity algorithm for searching on one-dimensional spaces. Finally, the vector of estimates for the *i*-th vector of DCT coefficients is built as
(5)$${\hat{s}}_{i}=\left[{\hat{s}}_{i,1},\ldots ,{\hat{s}}_{i,N_{i}}\right]\;i=1,\ldots ,n_{c}.$$

From these vectors of estimated coefficients, we can reconstruct each of the 8×8 blocks of coefficients in the frequency domain. Note that we need to pad zeros at the end of each block since the coefficients corresponding to the last $n_{b}-n_{c}$ vectors were not sent to the receiver. Thereby, the analog receiver provides the set of 8×8 blocks which are then converted back to the spatial domain using the IDCT to reconstruct the received image.

### 2.2. Digital Scheme

As introduced before and shown in Figure 1, we consider two different digital transmission schemes: one using a standard JPEG encoder, and another, named JPEGw/oEC, where the source encoder removes the redundant information in the same way as done in the analog system, i.e., by simply disregarding the vectors of coefficients corresponding to high frequencies in the DCT decomposition, and whose information is less valuable for the human visual perception system. Both encoders first quantize the input DCT coefficients by means of a quantization matrix. The quantization matrices used are the ones recommended by the JPEG standard [29] and are modified with the quality parameter from 1 to 99 to achieve different levels of compression. The quantized data is then encoded into a digital stream. For this matter, the JPEG encoder uses the entropic coder as defined in the JPEG standard [29]. The JPEGw/oEC encoder uses a simple fixed-lengh encoding scheme where each quantized data sample is encoded with the same number of bits. In this case, when increasing the compression level (i.e., reducing the quality parameter), a reduction of the number of bits of the encoded image is achieved because the maximum value of the quantized data is lower, thus the fixed number of bits needed to encode each sample is also lower.

The source-encoded binary sequence is then encoded using a forward error correction (FEC) encoder to obtain the sequence of channel encoded bits. In this work, we consider the same turbo code encoder as the one used in the LTE standard. Finally, the encoded bits are mapped into a QAM modulation and transmitted over the channel by the OFDM modulator. The possible combinations of coding rates and modulation orders are also taken from the LTE standard [30] (Table 7.2.3-1) and are reproduced in Table 1. We use the term channel quality indicator (CQI) as in LTE to refer to each possible combination of modulation order and coding rate.

## 3. Evaluation Methodology

In this section we explain the evaluation methodology and the different parameters of the systems. The considered analog and digital schemes are assessed by means of computer simulations and considering two different metrics: the SSIM index [24], and the transmission time, calculated by using the number of samples of the OFDM signal and the sampling rate of the LTE 10 MHz profile, 15.36 Msamples/s, as stated in Section 2. Note that the transmission time metric only considers the time that it takes to transmit the OFDM signal and not any other processing time such as the image compression time.

### 3.1. Analog JSCC Scheme

#### 3.1.1. System Input

We considered two different gray-scale (only gray-scale images are considered to simplify the assessment of the system, avoiding the complex evaluation of the three components present in color images) standard test images as shown in Figure 4: “Lenna” and “Gold Hill”. The “Lenna” image comprises two clearly differentiated parts, a woman face with a large level of detail and a blurred background. The “Gold Hill” image is a landscape picture with many focused objects with a lot of detail. The assessment of the system is done separately for each image.

#### 3.1.2. Analog Transmission Scheme

As it was stated in Section 2.1, to reduce the amount of transmitted data we disregard the vectors of coefficients corresponding to high frequencies in the DCT decomposition, and whose information is less valuable for the human visual perception system. There exists a large number of potential choices for the block division and posterior selection of the frequency coefficients. In this work, we have decided to set $n_{b}=4$ blocks with the following sizes: $N_{1}=1,N_{2}=3,N_{3}=12$, and $N_{4}=48$ (see Figure 3 for more details). This decision was motivated because this scheme matches to a multi-resolution decomposition with three levels, which was traditionally employed in image processing. Next, we considered two alternatives for the block selection step: (1) disregarding only the block $s_{4}$ corresponding to the highest frequencies, and (2) removing the coefficients of the last two blocks $s_{3}$ and $s_{4}$. Note that disregarding the vector $s_{4}$ had a minimum impact on the image quality, hence it will always be discarded. Disregarding $s_{3}$ may have a noticeable impact. However, for low SNR values, this impact was small, whereas the amount of transmitted data is significantly reduced. Henceforth, we refer to these schemes as “three blocks” and “two blocks”, respectively, and they are summarized in Table 2.

The two block scheme consisted of encoding $s_{1}$ with a different analog JSCC mapping depending on the SNR value of the received data, $s_{2}$ was transmitted uncoded, and the high-frequency data vectors, $s_{3}$ and $s_{4}$, were not transmitted. The different mappings considered for $s_{1}$ are summarized in Table 3. The three block scheme was similar to the 2 blocks scheme, with the only difference that the data in $s_{3}$ was transmitted uncoded, whereas $s_{4}$ was not transmitted.

Regarding the different mapping strategies for $s_{1}$, we considered the usage of different codes depending on the SNR value. This was motivated because $s_{1}$ corresponds to the lowest frequency components, and thus it contains the most valuable visual information. Therefore, as shown in Table 3, the level of redundancy is progressively reduced with the increase in SNR. We considered four intervals of SNR values corresponding to low, mid-low, mid-high, and high SNR as shown in Table 3. For very low SNR values, a linear repetition code was selected instead of the nonlinear version provided by spherical codes due to its smooth impact on the perceptual image quality. At low and medium SNR values, $s_{1}$ is encoded using spherical codes with an appropriate expansion factor. Finally, as the SNR increases, the benefits of the redundancy introduced by the codes are reduced and thus the data is transmitted uncoded. These schemes have been chosen after evaluating the system theoretically and practically with a trade-off between the efficiency based on the image quality at reception and the transmission time.

#### 3.1.3. Transmission and Channel Models

The analog JSCC symbols were packed into an OFDM frame and transmitted through the communications channel. We considered two different channel models:Additive white Gaussian noise (AWGN) channel: complex-valued AWGN noise with variance $\sigma^{2}$ is added to the OFDM signal.Flat Rayleigh channel: the OFDM signal was filtered with a channel coefficient generated from a complex standard Gaussian distribution, and complex-valued AWGN noise with variance $\sigma^{2}$ was added to the filtered signal.

The symbols at the OFDM modulator input have unitary variance for all the considered schemes. The OFDM modulator used a standard 1024-point inverse discrete Fourier transform (IDFT) for generating the time domain signal. This output signal was then normalized by the factor 600/1024, being 600 the number of data subcarriers used and 1024 the number of points of the IDFT, as explained before. This way, when adding AWGN noise with variance $\sigma^{2}$, the demodulated symbols in the receiver will have an SNR of $1/\sigma^{2}$. Based on this, we performed simulations considering 6 different values of $\sigma^{2}$ to obtain the following SNR values: 5, 10, 15, 20, 25, and 30 dB.

Note that the considered SNR values are actually the mean SNR values. For each channel realization, the “effective” SNR can be different. Taking this into account, to generate the analog transmit signal, the “effective” SNR for each channel realization is considered and not the mean SNR, i.e., the selected mapping for $s_{1}$ will depend on both the channel coefficient and the noise variance $\sigma^{2}$.

### 3.2. Digital Methodology

One of the main objectives of this work was to assess the proposed analog image transmission system compared to a digital one. The main idea employed to compare both systems, analog and digital, was to obtain an image with a similar quality (SSIM index value) for both systems. The first step was to transmit an image for a given channel realization using the analog system. The two different analog schemes, two blocks and three blocks (see Table 3), were considered, hence two transmissions were performed per input image and channel realization. Once the transmitted signal was received and both images were decoded, we calculated their SSIM index value.

Next, for the two received analog images (corresponding to the two block and three block schemes), digital images were source-encoded in such a way that the decoded image had an SSIM index value as close as possible to that of the image transmitted with the analog system. The SSIM was adjusted in this case by changing the quality parameter of the quantization matrix as explained in Section 2.2. Note that, for the JPEGw/oEC scheme, the images were encoded by discarding the same data vectors ($s_{4}$ and/or $s_{3}$) as the corresponding analog scheme, as shown in Table 3.

The source-encoded binary stream is then channel-encoded. This is done by using the optimum CQI value, which is the highest possible value ensuring an error-free transmission. Such a value was obtained by transmitting the image multiple times through the same channel realization for different CQI values and selecting the optimum one. Note that it was assumed that the CQI information is always decoded correctly at the receiver. The binary stream was mapped to QAM symbols and the OFDM signal is generated. Finally, the OFDM signal was transmitted using the same channel realization as in the transmission of the images with the analog system.

The assessment of the system performance is done based on the following figures of merit:Source size: this is the vector length of the binary vector of the JPEG or the JPEGw/oEC encoders.CQI: the optimum CQI value for an error-free transmission.Transmission time: the time to transmit the OFDM signal, as explained before, considering the sampling rate of the LTE 10 MHz profile.SSIM: The SSIM of the images encoded by the JPEG or JPEGw/oEC encoders.

## 4. Results

Simulations were performed as explained in Section 3 considering 100 different channel realizations. In this section we present and compare the obtained performance results of the considered systems using different figures of merit.

### 4.1. Visual Analysis

Comparing the quality of a compressed imaged with respect to the original one is often a challenging task, since an image may consist of several objects or parts with different levels of detail and relationships between them. Nowadays, the most used metric to perform this kind of comparison is the SSIM [24]. We used this figure of merit to analyze the visual quality of the images. However, note that due to the different nature of the disturbances added to the image by the analog and digital transmission schemes (i.e., AWGN noise vs. overall loss of details), the perceived quality of the images by the human eye may diverge from the quality indicated by the SSIM index.

As an example, Figure 5 shows the Lena image which is first transmitted with the analog system (left hand side) and, next, encoded with JPEG to obtain an SSIM index value as close as possible to that of the analog transmission (right hand side). Figure 6 and Figure 7 extend the comparison in Figure 5 for larger SSIM index values. It can be seen that the images transmitted with the analog system (left hand side of Figure 5, Figure 6 and Figure 7) are mainly altered with AWGN noise and the main image details are preserved for the three SSIM index values considered. The images transmitted with the JPEG scheme (right hand side of Figure 5, Figure 6 and Figure 7) are altered by the JPEG quantization quality parameter, which causes the loss of some details of the image.

The different nature of the alterations introduced causes that images with a similar SSIM index value may be perceived with a different quality by the human eye. In Figure 5 and Figure 6, it is possible to appreciate that the analog alterations are less aggressive than the digital ones to the human eye. As observed in Figure 7, these differences tend to vanish as the SSIM index value becomes larger, i.e., for a higher quality, but it is still visible.

### 4.2. Transmission Time

Figure 8, Figure 9, Figure 10 and Figure 11 show the transmission time with respect to the SNR for the two blocks and three blocks schemes, and for AWGN and Rayleigh channels. The transmission time of the analog scheme depends on the mapping applied to the data vector $s_{1}$ (see Figure Table 3), hence it depends on the effective SNR of the communications channel. As explained before, to obtain these results, once the image is received after being transmitted using the analog scheme, we transmit the image again through the same channel realization employing digital schemes and with an SSIM index value as close as possible to that corresponding to the analog transmission. Hence, the transmission time of the digital schemes will depend on the quality parameter of the quantization matrix and on the redundancy introduced by the turbo coder. As the SNR increases, the SSIM index value obtained after the analog transmission also increases, hence the number of source bits of the digital encoded images is also higher but, at the same time, the number of redundant bits introduced by the turbo coder becomes lower.

For the JPEG scheme, we can see in Figure 8, Figure 9, Figure 10 and Figure 11 that, for SNR values between 5 and 15 dB, the transmission time tends to decrease (the only exception is found in Figure 9). This effect is more noticeable when considering the two blocks analog scheme (Figure 8 and Figure 10). This is due to the reduction of the coding rate of the channel encoder as the SNR increases, while the improvement of the quality (SSIM) for the analog-encoded image is smaller in that SNR range. Thus, although the number of bits required at the output of the source encoder slightly increases for the digital schemes, the improvement of the OFDM modulation with the SNR is much higher, leading to both a reduction in the number of required channel symbols and a faster transmission. Conversely, for higher SNR values, the transmission time increases with the SNR. This effect is due to the quality improvement of the analog system in this SNR regime, increasing to a larger extent the number of bits required to encode the digital images, which cannot be compensated with the improvement of the OFDM modulation with the SNR. Additionally, the coding rate saturates at its maximum value (see Figure 12) and the amount of redundancy bits remains constant. More explanations about this behavior are provided in Section 4.3 and Section 4.4. Finally, the JPEGw/oEC scheme exhibits a similar behavior to that of the JPEG scheme, but the transmission times are worse because of the use of a simpler compression scheme.

The proposed analog scheme achieves its best results, in terms of transmission time, compared to the digital schemes for the two blocks case. Figure 8 and Figure 10 show that the transmission times of the analog scheme are very similar to those of the JPEG scheme, and even better for some SNR values. On the other hand, Figure 9 and Figure 11 show that when the three blocks scheme is considered, the transmission time is worse than that of the JPEG scheme, but still much better than that of the JPEGw/oEC scheme, especially for the Rayleigh channel model (see Figure 11).

Finally, we are interested in evaluating the performance of the proposed analog encoding system for a large set of images in order to ensure that the results obtained in the previous experiments for Lena and Gold Hill images can be extrapolated for the general case. We have selected eight typical images from a bank of test images (http://decsai.ugr.es/cvg/CG/base.htm) and measured the transmission times with the three considered encoding schemes. In particular, the images bridge, harbor, traffic, boat, mandrill, Barbara, Lena, and Gold Hill were chosen for this experiment. The transmission times obtained for the eigth images were averaged for each SNR value. Figure 13 shows the average times for the three encoding schemes considering the 2-blocks selection strategy and AWGN channels, whereas Figure 14 shows the same results for the case of Rayleigh channels. As observed, the obtained results agree with that of the previous experiments. Thus, the proposed analog JSCC scheme provides better results than the JPEGw/oEC for all the range of considered SNR values, and even outperforms the JPEG-based scheme in the high SNR regime.

### 4.3. SSIM

As explained in Section 3.2, when we transmit an image with the analog scheme, we generate images employing the proposed digital schemes with SSIM index values as close as possible to those obtained from the analog transmissions. However, since the analog and digital schemes are essentially different, it may be impossible to obtain the exact same SSIM index values for both analog and digital schemes under the same channel conditions. To illustrate this, we show in Figure 15, Figure 16, Figure 17 and Figure 18 the SSIM index values of the considered images with respect to the SNR for the two block and three block schemes, considering AWGN and Rayleigh channels.

As expected, the quality achieved with the three blocks scheme (see Figure 16 and Figure 18) is better than that obtained with the two blocks schemes for both channel models, AWGN and Rayleigh. For low SNR values, the SSIM index value of the analog scheme diverges from that of the digital schemes, and this divergence is larger for the AWGN channel than for the Rayleigh channel. However, at this point, it is worth remarking that digital images with larger SSIM index values can be perceived with a worse quality by the human eye than those with a smaller SSIM index value (e.g., see Figure 5). Therefore, the observed divergence (less than 0.1 in terms of SSIM) is considered acceptable for our comparison purposes. On the other hand, for higher SNR values, the SSIM index values of the analog and digital schemes becomes approximately the same.

Note that the quality of the image transmitted with the analog scheme may decrease without any limit as the noise becomes larger. However, for the digital schemes, the quality has a minimum value. Therefore, the comparison of the system is not totally fair in the low SNR regime. This is the motivation to select 5 dB as the lower value for the SNR.

### 4.4. CQI and Source Size

Figure 12 shows the CQI of the data transmitted with the digital schemes with respect to the SNR for the case of a Rayleigh channel. The difference between JPEG and JPEGw/oEC is because the amount of data transmitted with JPEGw/oEC is larger, hence increasing the probability of suffering an error, thus reducing the optimum CQI value for that cases. The results corresponding to the AWGN channel are very similar to those shown in Figure 12 and were not included.

Figure 19 and Figure 20 show the amount of bits to be transmitted (once the images are encoded using the digital schemes) with respect to the SNR for the case of the 2 blocks and 3 blocks and considering AWGN channel. Notice that the amount of bits depends on the image quality. When the SNR of the link increases, the analog transmissions produce images with a higher SSIM index value, hence the amount of bits required to encode the images with that quality also increases.

## 5. Image Transmission: Analog or Digital?

In this section we summarize the advantages and disadvantages, in terms of image quality and transmission time, of the proposed analog system compared to the considered digital alternatives.
Visual degradation: as shown in Figure 5, Figure 6 and Figure 7, the visual quality perception is different for the images transmitted with the proposed analog system or encoded with JPEG. The analog system quality is better to the human eye because the received image is distorted with AWGN-like noise, preserving the details in the original image. However, in the JPEG-encoded image, some of the details are removed from the image as a consequence of the quantization process.Computational complexity: the complexity of the proposed analog scheme is basically determined by the decoding operation at the receiver, since the computational cost of the mapping operation is negligible, whereas the DCT and OFDM are linear operations which can be carried out efficiently in a practical implementation. The complexity of the decoder is minimum for uncoded transmissions and, although is higher for the case of spherical codes, the corresponding ML decoder can be implemented using efficient search algorithms for one-dimensional space.On the other hand, the highest complexity operation in the two digital schemes corresponds to the iterative decoding required by the turbo codes. In practice, there exist efficient implementations for the turbo decoder, but their iterative nature leads to a computational cost larger than that of the analog decoder. In addition, the communication delay will also be larger due to the need of receiving a block of bits long enough to proceed with the decoding. As a conclusion, the obtained results show that the analog scheme is able to provide a similar performance to that of the digital one based on JPEG but with lower complexity and delay.Metadata transmission (overhead): as explained above, the design of the analog and digital schemes should involve an essential step to obtain an efficient representation of the image contents with the minimum amount of coefficients to be encoded. In this work, we decided to apply a compression scheme based on a frequency block division and a posterior selection of the coefficient blocks to be transmitted. The same scheme is applied to all the 8×8 pixel-blocks of the image. This simple strategy presents an important advantage with respect to other dynamic approaches since the amount of required metadata becomes negligible. Moreover, for transmitting similar images we could consider preseting the metadata at both transmitter and receiver, hence obtaning a fully analog system.An alternative procedure would be to adapt the number of blocks and their size, as well as the number and position of the disregarded blocks, depending on the properties of each 8×8 input block. This strategy improves the compression operation at the expense of using a large amount of metadata for the image reconstruction, hence increasing the complexity of the transmission scheme; the total amount of data to be transmitted; and the probability of an erroneous transmission, since the metadata information becomes indispensable.Fixed transmission rate and low delay: considering images with the same size and a constant SNR value, the proposed analog system exhibits a fixed transmission rate (i.e., the time required to transmit each image). Conversely, digital schemes adapt the transmitted rate depending on the channel conditions, as detailed in Section 4.4. Moreover, retransmissions may be needed if the received data contains errors, thus increasing the actual transmission time.System optimization: both the proposed analog JSCC scheme and the two digital systems should be optimized depending on the channel conditions with the aim of providing the best possible performance. In the case of the proposed analog scheme, the optimization is limited to the choice of the expansion factor and the mapping strategy as a function of the mean SNR value. When using spherical codes, it is also possible to change the α parameter as described in Section 2.1, although its impact on the image quality is negligible for the considered range of SNR values. In the case of the digital systems, however, it is required to select an appropriate value for the CQI parameter, which ultimately determines the rate of the channel encoder and the number of modulation levels. The CQI value must be chosen to ensure an error-free transmission of the encoded bits at the highest possible speed and is calculated independently for each channel realization.When perfect CSI is available at the transmitter, the optimization procedure becomes similar for both transmission schemes. However, this assumption can be too much optimistic for a large range of scenarios where the feedback is limited due to multiple factors, in such manner that the transmitter must deal with inaccurate channel information. In such a case, considering a CQI value lower than the optimal one will lead to an increment of the transmission time, whereas the received image will preserve its original quality. Conversely, using a CQI value higher than the optimal one will have fatal consequences since the transmission errors will make the image reconstruction impossible. Hence, the information corresponding to that block would need to be retransmitted, causing a severe overhead and delay which is not acceptable, for example, in real-time applications. For analog transmissions, the impact of using too pessimistic parameters has a similar impact as in the digital case. However, the other situation (too optimistic parameters) is clearly favorable for the analog transmission strategy since the information is received with a gradual performance loss, hence making the image reconstruction possible.

## 6. Conclusions

In this paper, an analog JSCC system designed for the transmission of still images was proposed. The performance of the proposed system was compared to that of two digital alternatives which differ in the source encoding operation: JPEG and JPEGw/oEC, respectively, whereas both employ an optimized channel encoder-modulator tandem. Apart from a visual comparison, the figures of merit considered in the assessment were the SSIM index and the time required to transmit an image.

The obtained results showed that the proposed analog system exhibits a performance similar to that of the digital scheme based on JPEG compression, with a noticeable better visual degradation to the human eye, a lower computational complexity, and a negligible delay. These results confirmed the suitability of analog JSCC for the transmission of still images in scenarios with severe constraints on the power consumption, computational capabilities, and for real-time applications. Additionally, the proposed analog scheme does not required to transmit any metadata information, hence the analog data symbols can always be processed at the receiver. Conversely, digital systems relying on adaptive modulation and coding schemes have to transmit the information about the selected CQI to the receiver. In case such information is corrupted, all transmitted data is lost and cannot be processed at the receiver.

## Figures and Tables

**Figure 1 sensors-19-02932-f001:**
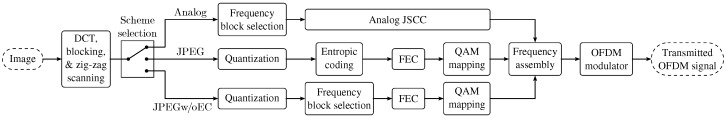
Block diagram of the transmission scheme.

**Figure 2 sensors-19-02932-f002:**
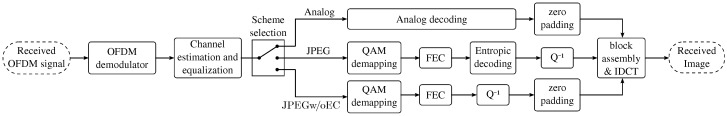
Block diagram of the reception scheme.

**Figure 3 sensors-19-02932-f003:**
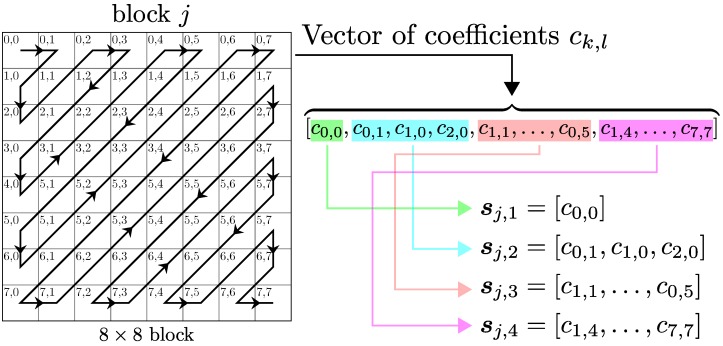
Graphical representation of the division applied to the *j*-th block of 8×8 discrete cosine transform (DCT) coefficients considering $n_{b}=4$ sub-blocks, and the way of building the corresponding vectors of coefficients with different levels of detail for the image.

**Figure 4 sensors-19-02932-f004:**
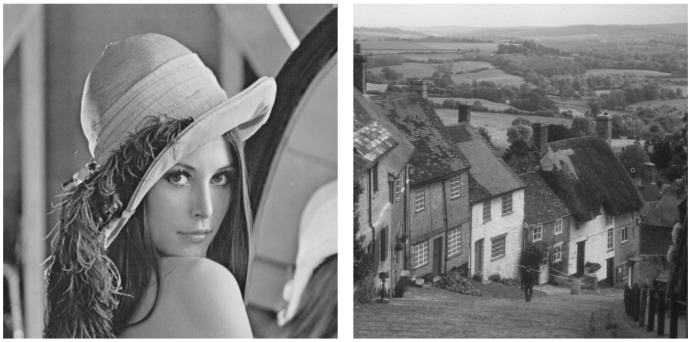
Input images: Lena and Gold Hill. Size: 512×512 pixel each image.

**Figure 5 sensors-19-02932-f005:**
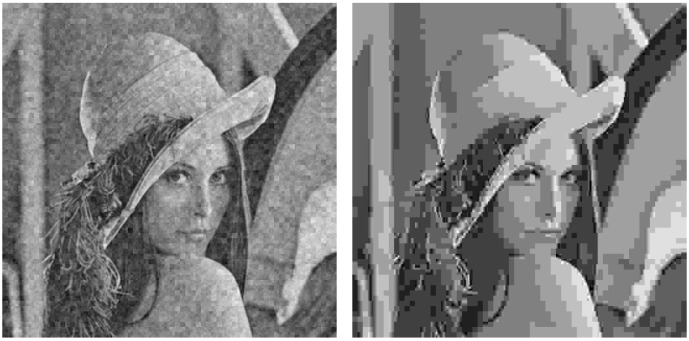
Output images. **Left hand side**: Lena, analog transmission, 0.59 structural similarity (SSIM). **Right hand side**: Lena, JPEG encoding, 0.64 SSIM.

**Figure 6 sensors-19-02932-f006:**
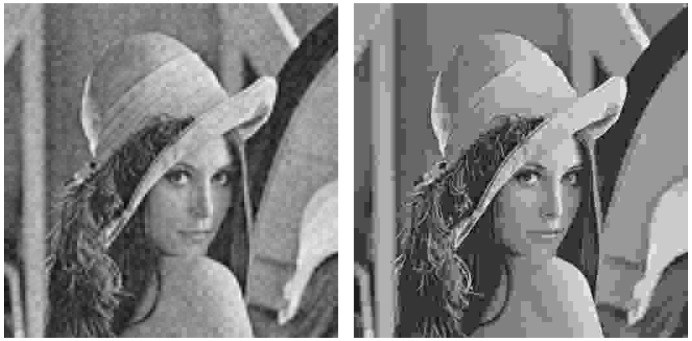
Output images. **Left hand side**: Lena, analog transmission, 0.70 SSIM. **Right hand side**: Lena, JPEG encoding, 0.72 SSIM.

**Figure 7 sensors-19-02932-f007:**
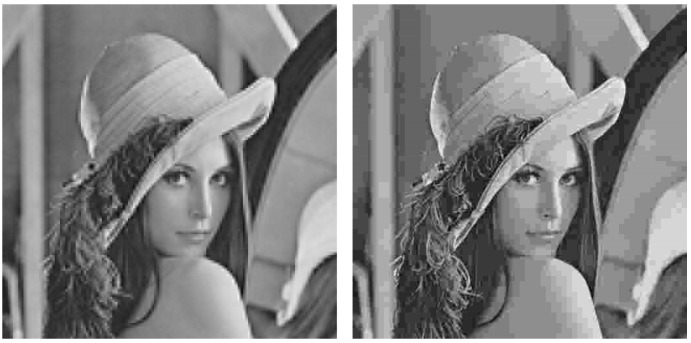
Output images. **Left hand side**: Lena, analog transmission, 0.80 SSIM. **Right hand side**: Lena, JPEG encoding, 0.78 SSIM.

**Figure 8 sensors-19-02932-f008:**
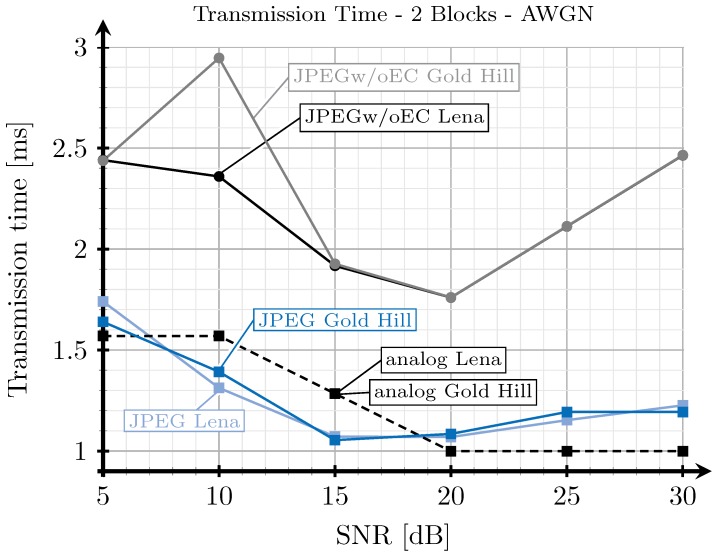
Transmission time vs. the signal to noise ratio (SNR) for two blocks and additive white Gaussian noise (AWGN) channel.

**Figure 9 sensors-19-02932-f009:**
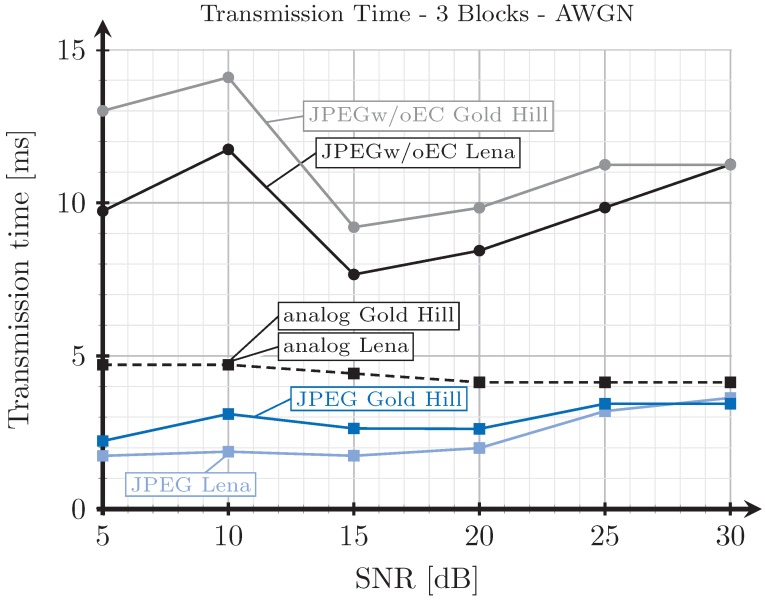
Transmission time vs. SNR for three blocks and AWGN channel.

**Figure 10 sensors-19-02932-f010:**
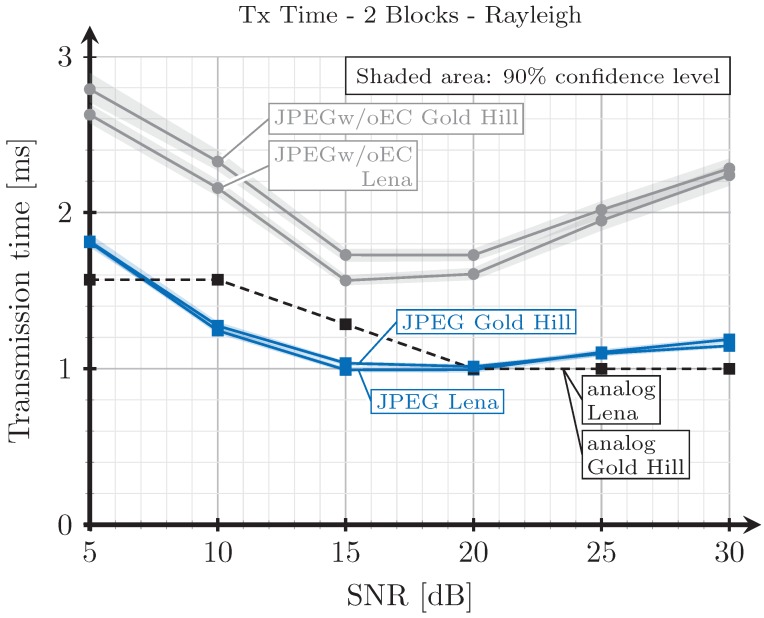
Transmission time vs. SNR for two blocks and Rayleigh channel.

**Figure 11 sensors-19-02932-f011:**
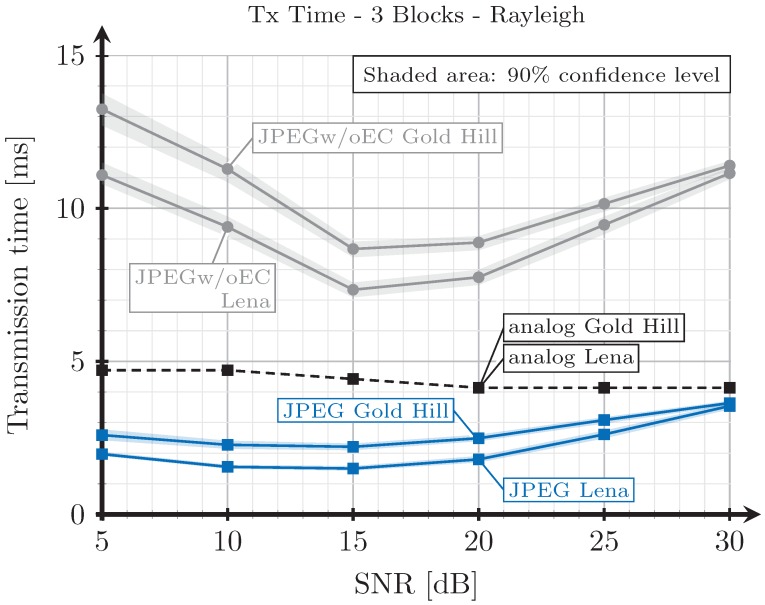
Transmission time vs. SNR for three blocks and Rayleigh channel.

**Figure 12 sensors-19-02932-f012:**
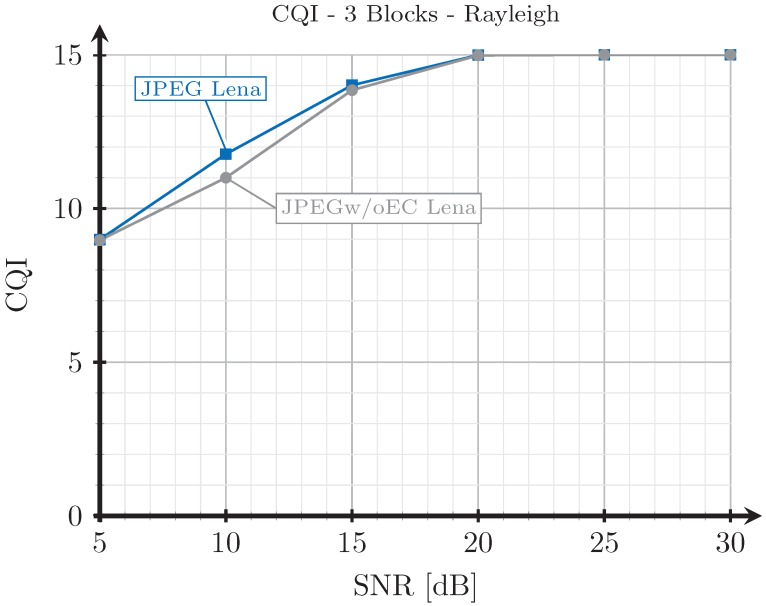
Channel quality indicator CQI vs. SNR for 3 blocks and Rayleigh channel.

**Figure 13 sensors-19-02932-f013:**
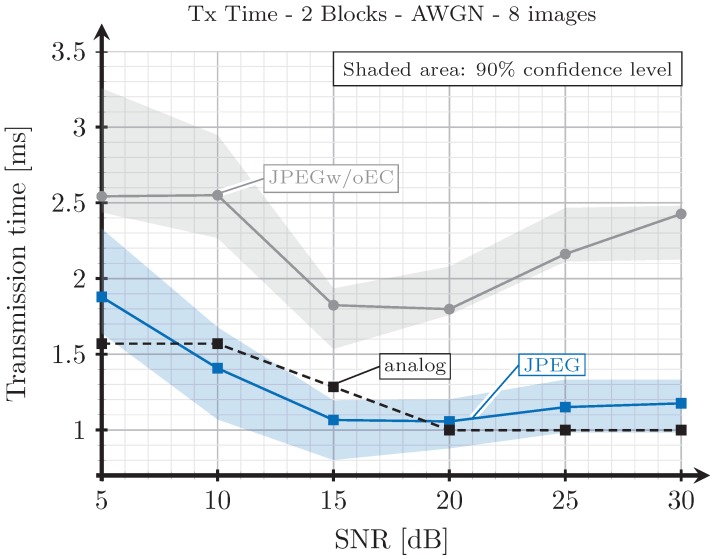
Average transmission time (considering 8 images) vs. SNR for two blocks and AWGN channel.

**Figure 14 sensors-19-02932-f014:**
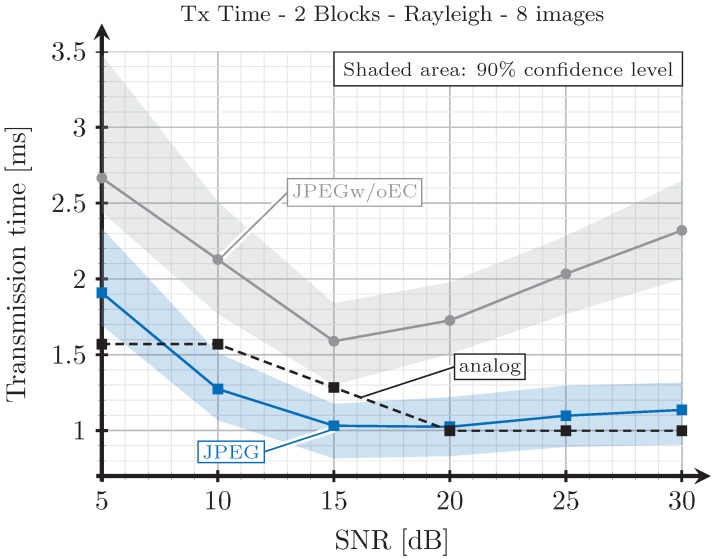
Average transmission time (considering 8 images) vs. SNR for two blocks and Rayleigh channel.

**Figure 15 sensors-19-02932-f015:**
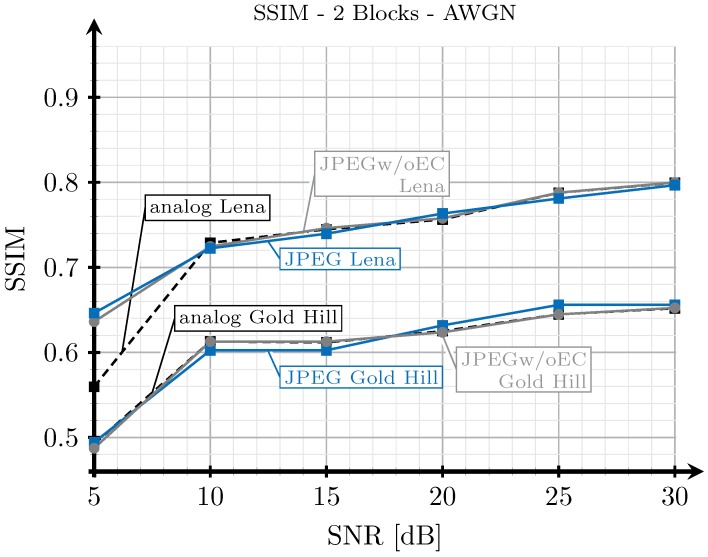
SSIM vs. SNR for two blocks and AWGN channel.

**Figure 16 sensors-19-02932-f016:**
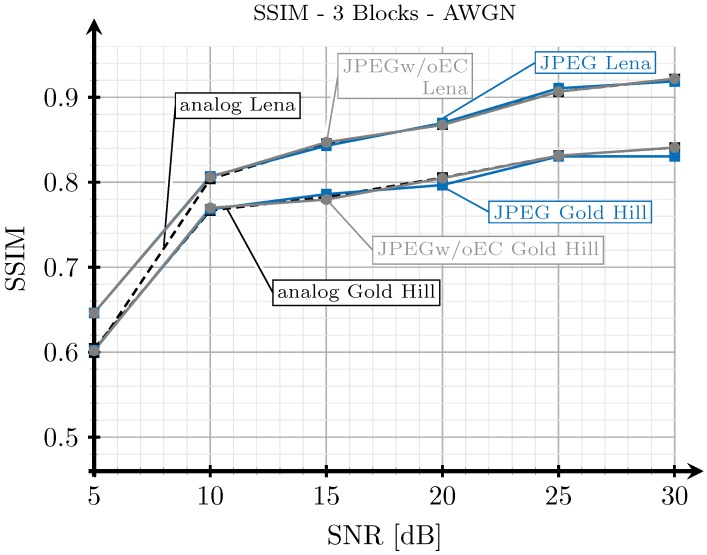
SSIM vs. SNR for three blocks and AWGN channel.

**Figure 17 sensors-19-02932-f017:**
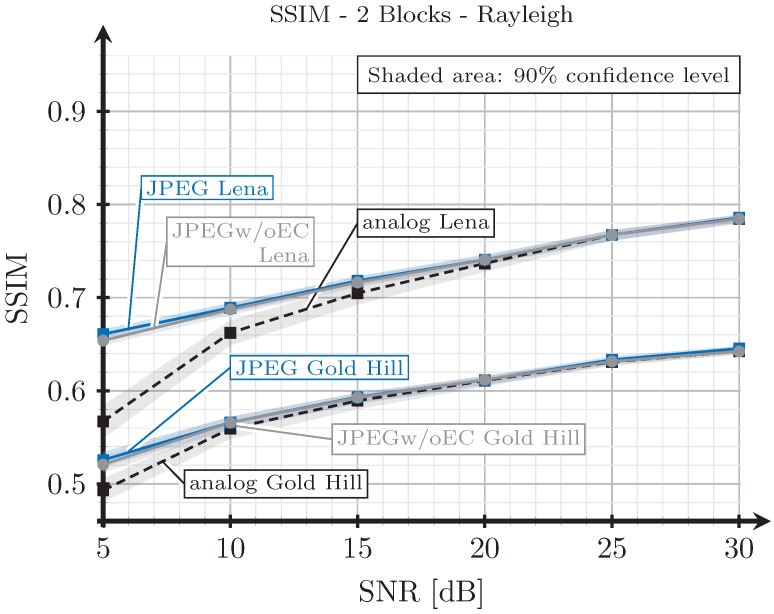
SSIM vs. SNR for two blocks and Rayleigh channel.

**Figure 18 sensors-19-02932-f018:**
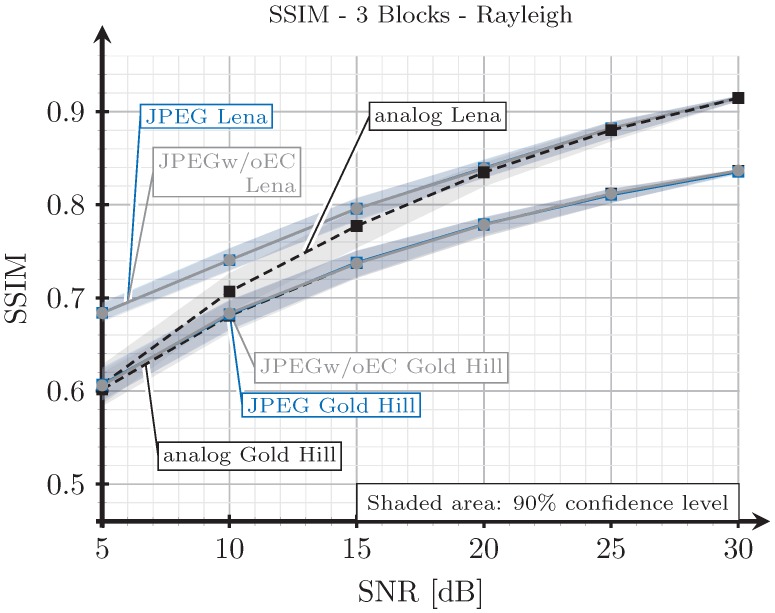
SSIM vs. SNR for three blocks and Rayleigh channel.

**Figure 19 sensors-19-02932-f019:**
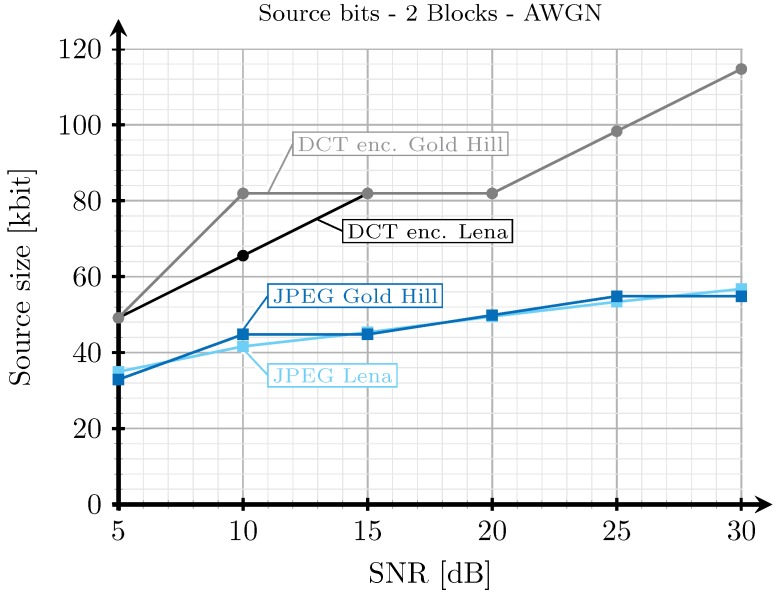
Source size vs. SNR for two blocks and AWGN channel.

**Figure 20 sensors-19-02932-f020:**
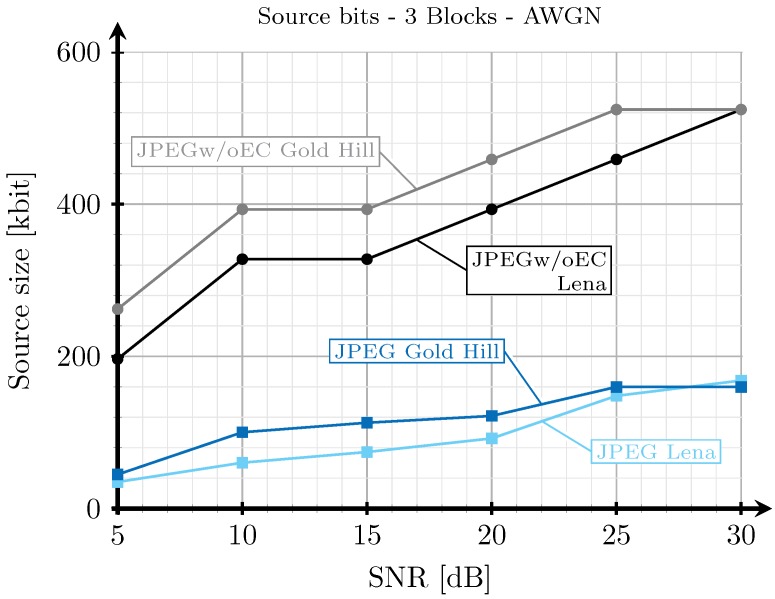
Source size vs. SNR for three blocks and AWGN channel.

**Table 1 sensors-19-02932-t001:** Possible combinations of coding rates and modulation orders (**CQI!**s) used in the digital schemes.

CQI Index	Constellation	Code Rate × 1024	Efficiency
1	4-QAM	78	0.1523
2	4-QAM	120	0.2344
3	4-QAM	193	0.3770
4	4-QAM	308	0.6016
5	4-QAM	449	0.8770
6	4-QAM	602	1.1758
7	16-QAM	378	1.4766
8	16-QAM	490	1.9141
9	16-QAM	616	2.4063
10	64-QAM	466	2.7305
11	64-QAM	567	3.3223
12	64-QAM	666	3.9023
13	64-QAM	772	4.5234
14	64-QAM	873	5.1152
15	64-QAM	948	5.5547

**Table 2 sensors-19-02932-t002:** Analog transmission schemes.

Scheme	Analog Mappings for the Data Vectors			
	$s_{1}$	$s_{2}$	$s_{3}$	$s_{4}$
two blocks	depending on the SNR (see Table 3)	uncoded	do not transmit	do not transmit
three blocks	depending on the SNR (see Table 3)	uncoded	uncoded	do not transmit

**Table 3 sensors-19-02932-t003:** Mappings for $s_{1}$ (see Table 2).

SNR (dB)	Analog Mapping
<6	Repetition factor 4
6 to 11	Spherical code factor 4
11 to 16	Spherical code factor 2
>16	Uncoded

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
