# Peer review of "Transmission of Still Images Using Low-Complexity Analog Joint Source-Channel Coding"

_sensors, 2019, doi:10.3390/s19132932_

Round 1

Reviewer 1 Report

The title of this paper looks analog circuit for image encoder in the context of image compression. However, the proposed system is aimed at analog transmission or joint source-channel coding. Please consider a suitable title.

Is the process of DCT for a digital image processing? Please notify it. Also, blocking and zigzag scanning processing can be included in Figs 1,2.

There are several options for DCT in JPEG, such as floating-point precision, integer-precision with LLM, and integer-precision with AAN. What type of DCT did you use?

The encoding method of DCT is confusing. Another name is better, such as JPEG without entropy coding.

Line 130, Turbo Codes-> turbo codes: Besides, there is a fast JPEG compression library, named JPEG-Turbo. This is a little confusing; thus, the following sentence is a candidate for modification.

we consider the standard JPEG and Turbo Codes as representative components for the digital system

->

we consider the standard JPEG for image coding and turbo codes for channel coding as representative components for the digital system.

How to simulate transition time? Does this time include image compression time? Please notify this point.

Reviewer 2 Report

The manuscript reported an investigation of an analogue joint source-channel coding method for still image transmission via wireless link based on the orthogonal frequency-division multiplexing (OFDM) modulation with assumed signal to noise ratio (SNR) where the noise was characterised by either additive white Gaussian noise (AWGN) and Rayleigh distributions, respectively. The presented method offered an alternatively approach to picture transmission over wireless channel, which has lower computational complexity compared with classical source-channel coding theory with digital coding of images accompanied by forward error correction. The structure similarity index (SSIM) was used as a picture quality measure which was made "constant" or comparable in the performance evaluations in terms of the visual picture quality and transmission time. The system level descriptions of the method as shown in Figs. 1 and 2 are clear and insightful, in comparison with digital coding methods as benchmarks. The manuscript is easy to read and to follow.

Specific comments were provided for the authors' consideration.

The presentation should be made consistent. For instance, "joint source-channel coding" was used in the abstract, while "combined source channel coding" was used in the key words.

The first paragraph of Introduction section (Lines 16-26) made a number of statements and assertions which were not supported by evidence or references. The statement was not entirely accurate, i.e., "...an adequate source encoder is employed to remove the spatial and statistical redundancy to obtain an efficient digital representation of the image." See, e.g., [i] with regard to statistical (including source coding, spatio-temporal and inter-scale) and psychovisual (or perceptual) redundancies contained in natural images or visual signals.

It is known from [19] of the manuscript that the SSIM favours images with blurring distortion to those with blocking artefacts (i.e., SSIM values being equal or comparable, images with blurring distortion visually look less objectionable than those with blocking artefacts in the mid to low coding bitrate ranges.) It begs the question as to why the SSIM was chosen as the picture quality measure and made to be constant or comparable in the performance evaluation. In other words, knowing the limitations of the SSIM, using SSIM as figure of merit was not fully justified for performance comparison.

Lines 87-93: further clarification or discussion is required to elaborate that the analogue JSCC used fixed grouping and quantisation while visual contents dependency is a key consideration for design of advanced image coding methods.

Lines 167-169: How to assemble a comparable symbol vector size for all three schemes feeding into OFDM?

Lines 200-207: Mathematical description could be provided for the stacking process. Citation of Fig.3 here seemed to be misleading, since Fig.3 only defined blocks instead of the block stacking results. It "resembles" subband coding" [ii].

 Index "i" in (1) requires clarification, since Fig.3 showed 4 blocks, and "i" in (1) can be more than 4, if that would be the case.

 Regarding (3), s or s is unknown to the receiver, right?

More test images than the two images (Lena and Gold Hill) should be used in the investigation.

Line 2: "...compared with..."

Line 13: "...Internet..."

Line 17: "in the context of the Internet of things (IoT)."

Line 357: obtaining the SSIM value [19 ] .

References

[i]   H.R. Wu and K.R. Rao, (eds.) Digital Video Image Quality and Perceptual Coding, CRC Press, 2006.

[ii] R.J. Clarke, Digital Compression of Still Images and Video, Academic Press, 1995.

.
